# Novel In Silico Insights into Rv1417 and Rv2617c as Potential Protein Targets: The Importance of the Medium on the Structural Interactions with Exported Repetitive Protein (Erp) of *Mycobacterium tuberculosis*

**DOI:** 10.3390/polym14132577

**Published:** 2022-06-25

**Authors:** Margot Paco-Chipana, Camilo Febres-Molina, Jorge Alberto Aguilar-Pineda, Badhin Gómez

**Affiliations:** 1Centro de Investigación en Ingeniería Molecular—CIIM, Universidad Católica de Santa María, Urb. San José s/n—Umacollo, Arequipa 04000, Peru; 74252022@ucsm.edu.pe (M.P.-C.); c.febresmolina@uandresbello.edu (C.F.-M.); jaguilar@ucsm.edu.pe (J.A.A.-P.); 2Doctorado en Fisicoquímica Molecular, Facultad de Ciencias Exactas, Universidad Andrés Bello, Santiago 8320000, Chile

**Keywords:** tuberculosis, Erp, Rv1417, Rv2617c, molecular dynamics

## Abstract

Nowadays, tuberculosis is the second leading cause of death from a monopathogenic transmitted disease, only ahead of COVID-19. The role of exported repetitive protein (Erp) in the virulence of *Mycobacterium tuberculosis* has been extensively demonstrated. In vitro and in vivo assays have identified that Erp interacts with Rv1417 and Rv2617c proteins, forming putative transient molecular complexes prior to localization to the cell envelope. Although new insights into the interactions and functions of Erp have emerged over the years, knowledge about its structure and protein–protein interactions at the atomistic level has not been sufficiently explored. In this work, we have combined several in silico methodologies to gain new insights into the structural relationship between these proteins. Two system conditions were evaluated by MD simulations: Rv1417 and Rv2617c embedded in a lipid membrane and another with a semi-polar solvent to mimic the electrostatic conditions on the membrane surface. The Erp protein was simulated as an unanchored structure. Stabilized structures were docked, and complexes were evaluated to recognize the main residues involved in protein–protein interactions. Our results show the influence of the medium on the structural conformation of proteins. Globular conformations were favored under high polarity conditions and showed a higher energetic affinity in complex formation. Meanwhile, disordered conformations were favored under semi-polar conditions and an increase in the number of contacts between residues was observed. In addition, the electrostatic potential analysis showed remarkable changes in protein interactions due to the polarity of the medium, demonstrating the relevance of Erp protein in heterodimer formation. On the other hand, contact analysis showed that several C-terminal residues of Erp were involved in the protein interactions, which seems to contradict experimental observations; however, these complexes could be transient forms. The findings presented in this work are intended to open new perspectives in the studies of Erp protein molecular interactions and to improve the knowledge about its function and role in the virulence of *Mycobacterium tuberculosis*.

## 1. Introduction

Before the COVID-19 pandemic, tuberculosis (TB) was considered the foremost cause of death from an infectious disease worldwide [1,2]. In addition, the pandemic is expected to increase mortality by more than 20% globally [3,4]. According to the latest data from the World Health Organization (WHO) regarding TB, it was estimated that in 2020, around 10 million people were infected with TB, of which 11% were children [5]. Of the total infected, 15% died of causes derived from TB [6]. Even more worrying is that it is estimated that at least a quarter of the world’s population has the disease and a large percentage are asymptomatic (LTBI, latent tuberculosis infection). Various reports indicate that LTBI can remain dormant for decades and return to active disease in almost 10% of infected people [6,7,8].

TB is a chronic disease caused by the intracellular pathogen *Mycobacterium tuberculosis* (*Mtb*) [5,8,9]. Despite being a preventable and curable disease, TB is associated with high rates of morbidity and mortality [9,10]. The ignorance of the immune response in the progression from the latent to the active form in TB, in addition to the different mechanisms of the bacteria to survive in hostile conditions, has resulted in an increase in deaths in recent years [6,11]. Furthermore, an increase in the incidence of TB is expected throughout 2022 due, among several reasons, to the drug resistance developed by *Mtb* [12]. Four drugs are commonly used in conventional TB treatment: isoniazid, rifampin, pyrazinamide, and ethambutol or streptomycin, which are known as first-line drugs [13]. Unfortunately, the WHO has identified strains resistant to these drugs in all the countries studied [5]. Approved second-line drugs for multidrug-resistant TB (MDR-TB) and drug-resistant TB (XDR-TB) treatments have shown 50% efficacy (ethionamide, capreomycin and kanamycin) [14]. The most effective drugs, such as ofloxacin and norfloxacin, are expensive and require intense chemotherapy and continuous monitoring, which affects the quality of life of the sufferer [13,14].

Considerable efforts have been made in recent years to understand the pathogenesis, virulence, and genomics of *Mtb* [9,15,16,17,18]. The existence of different genes linked to virulence that allow them to survive and persist in hosts has been demonstrated. Research by Madacki et al. [19] mentions that *Mtb* does not produce a single dominant virulence factor responsible for causing the disease but induces it through several groups of factors that intervene in the different stages of pathogenesis. In this sense, in vitro cell culture assays, and in vivo, have demonstrated the determinant role of the exported repetitive protein (Erp) in the virulence factor of *Mtb* [15,20,21]. Berthet et al. showed that a disruption in Erp gen created *Mtb* mutant strains that were rapidly cleared from the lungs of infected animals [22]. However, its exact role in Mtb virulence mechanisms remains hidden. However, not all virulence genes have been studied, classifying them as unknown, including some transmembrane proteins [23]. To clarify the role of Erp, Klepp et al. carried out a search for Erp-binding proteins and found that the proteins Rv1417 and Rv2617c interacted with it [24]. Despite not knowing the locations of Rv1417 and Rv2617c, the authors mentioned that both could be located in the cytoplasmic membrane, while Erp is in the cell wall. In addition, there are studies indicating that Erp is found in both intracellular and extracellular regions [21,22,25]. Klepp et al. also reported that interactions between these three proteins produce a multimeric structure on the surface of the membrane, although it is not specified whether this may influence the virulence capacity of Erp [24]. Recent research showed that Rv2617c is relevant for the arrest of *Mtb*-induced phagosome maturation and is involved in the oxidative stress resistance mechanism [23].

The lack of knowledge of the molecular interactions between *Mtb*-secreting proteins and their relationship with infection and virulence factors may undermine the development of new anti-TB drugs with great potential. Erp is one of the main proteins involved in *Mtb* virulence [15,19,20,21,22]. Under in vitro and in vivo conditions, Erp has been shown to bind to the Rv1417 and Rv2617c proteins [24]. The possible interruption of these interactions could alter its function and decrease its virulence.

The aim of this work is to explore the conformational dynamics of Rv1417 and Rv2617c proteins under membrane and solution conditions on a nanosecond timescale. Furthermore, their interactions with the Erp protein have been investigated. MD simulations and various in silico techniques are carried out to identify important residues involved in protein–protein interactions. The structural insights gained from this study shed light on the binding modes and active sites of the analyzed proteins.

## 2. Computational Details

### 2.1. System Preparation

To explore the protein–protein interactions, we selected three *Mtb* protein structures. Sequences and structures of the exported repetitive protein, Erp (Rv3810) [26], uncharacterized protein Rv1417 [27], and probable transmembrane protein Rv2617c [28] of *Mtb* bacterium were retrieved from the Uniprot and AlphaFold databases [29,30,31].

#### Erp, Rv1417 and Rv2617c Proteins

Prior to preparing the protein systems for the MD simulations, the initial 3D structures were obtained in two ways. First, the structures predicted by the AlphaFold methodology were retrieved from the server database and used as initial models. Second, based on the amino acid sequences, the structures were modeled using the I-TASSER [32,33,34] server. The models with the best C score were optimized with the ModRefiner server [35]. AlphaFold models (AFMs) showed well-defined α-helix structures that could be embedded in lipid membranes, while I-TASSER models (ITMs) showed globular structures. Due to those configurational differences, molecular systems were prepared under different conditions: the first with an explicit lipid membrane (AFMs) and the second in a semi-polar environment (ITMs).

For the protein membrane, explicit water molecules were considered as solvent. The Rv1417 and Rv2617c proteins (RvPs) were embedded in a dipalmitoylphosphatidylcholine (DPPC) bilayer with 512 and 509 lipids, respectively. A 128-DPPC bilayer with 64 lipid molecules for each layer was replicated four times (twice in the x and y directions) to obtain the membrane model. The InflateGRO methodology was used for the incorporation of proteins into the lipid membrane [36]. Only protein systems were constructed using ethanol as a semi-polar solvent (dielectric constant = 23.58 at 309.65 K [37]). In both the membrane and semipolar systems, the Erp systems were prepared considering the isolated protein.

### 2.2. MD Simulations

Molecular dynamics (MD) simulations were performed in Gromacs 2020.3 [38,39] with the OPLS-AA force field. In all systems, vacuum energy minimization was performed with the steepest descent algorithm with a maximum of 50,000 steps. In the membrane systems, once the protein–membrane complexes were obtained, they were located in the center of a cubic box with a distance of 1.0 nm between the surface of the system and the border of the box on the *z* axis. TIP4P water model molecules were used to solvate the systems [40]. Next, we proceeded to another energy minimization with a maximum of 50,000 steps using the DPPC parameters for the lipid bilayer obtained from the work of Peter Tieleman et al. For the semi-polar systems, the proteins were placed in the center of a cubic box in which the distance from the system to the borders of the box in all directions was 1.3 nm. These structures were solvated using ethanol molecules [41]. To neutralize the systems and mimic physiological conditions, Na${}^{+}$ and Cl${}^{-}$ were added to obtain an ionic strength of 150 mM with a neutral total net charge. The main simulation conditions of the equilibrium and production phases for the membrane and semipolar systems are shown in Table 1. For the analyses, all trajectories were saved every 15 ps.

### 2.3. Molecular Docking Calculations

Once the minimum energy structures were obtained, several docking calculations were performed to analyze the multiple interactions. A first docking was performed between RvPs and Erp protein to obtain the main interacting residues in the protein–protein complexes. For this, and based on complementary forms, the PatchDock server was used to obtain the complexes [42,43]. The best structures were refined on the FireDock server, which allowed re-scoring the solutions based on an energy function [44,45]. In membrane protein coupling, a second molecular docking was performed to evaluate which would be the best structure to calculate the interaction complexes. For this, the MemDock server was used [46].

### 2.4. Structure and Data Analysis

The statistical results, root mean square deviation (RMSD), root mean square fluctuation (RMSF), radii of gyration (RG), solvent-accessible surface area (SASA), hydrogen bonds (HB), binding free energies (BFE), structures, trajectories and B-factor maps were obtained using the Gromacs modules. Analysis of the structure properties was performed using the MD trajectories of the last 200 ns of each simulation and then visualized using the Visual Molecular Dynamics (VMD) [47] and the UCSF Chimera v.1.14 [48] softwares. Graphs were plotted using the XMGrace software [49]. Two-dimensional (2D) representations of the electrostatic and hydrophobic interactions were constructed using the LigPlot [50] program. Electrostatic potential (ESP) surfaces in the molecular mechanics framework were calculated with the APBS (Adaptive Poisson Boltzmann Surface) software v.1.4.1, [51], and the pqr input was created in the PDB2PQR [52] server. Free energy landscape (FEL) maps were used to visualize the energy associated with protein conformation of the different models during MD simulations. These maps were plotted using the *gmx sham* module, while the RMSD and RG were considered as the atomic position variables with respect to their mean structure. Figures related to the latter were constructed using Wolfram Mathematica 12.1 [53].

## 3. Results and Discussion

Experimental evidence has shown that RvPs interact with the Erp protein, which is one of the important virulence factors of *Mtb* [24]. These interactions have been linked to bacterial replication in a mouse model of infection as well as to the prevention of bacterial injury by oxidative stress [23]. Despite the important role these proteins appear to play, how they interact remains unknown. Moreover, although they have been identified as probable membrane proteins, their exact localization has not been established, making them more difficult to study.

To address these issues, MD simulations and docking analyses were performed to evaluate protein–protein interactions. For this purpose, RvPs structures were embedded in a DPPC bilayer membrane in an aqueous solution, and the final MD complexes were docked with the isolated Erp protein. Finally, electrostatic potential and dynamic pocket analyses were performed to identify the interaction sites of potential drug targets.

### 3.1. Structure Characterization

The characterization of protein structure is an important key to understanding protein functions. Prior to MD simulations, analyses of the topology of transmembrane segments and intrinsically disordered regions (IDRs) were performed to gain insight into the flexibility domains as well as the position and orientation of proteins in the lipid bilayer. The membrane topologies of proteins play an important role in their physiological function and are crucial in the recognition of novel therapeutic targets [54,55,56]. The protein orientations were obtained using the DeepTMHMM server, Ref. [56], which uses deep learning methods to predict protein topologies. The calculated percentage values represent the probability of residue location in the different regions of the membrane systems. Moreover, regions of disorder provide functional advantages to proteins in cell signaling processes, including promiscuous interactions with different targets [57,58,59,60]. Disorder predictions were performed on the IUPred2A server using the IUPred2 and ANCHOR2 algorithms [61,62]. IUPred2 predicts the probability that certain residues are part of disordered regions. ANCHOR2 values are the probability that such residues are part of disordered binding regions (DBRs). Probability scores range from 0 (ordered or non-binding regions) to 1 (disordered or binding regions).

The results show important differences with respect to previous analyses on the orientation of RvPs. Klepp et al. mention that Rv2617c has three transmembrane helices, and the probability that the C-terminal domain is oriented outward is about 82%. Using the improved TMHMM server (inner plots in Figure 1A–C), the helical structure prediction for Rv2617c was four α-helices (V24-L39, Q70-P90, I92-P112, and I117-A135), which is in agreement with the AFMs structures. Moreover, the N- and C-terminal domains of RvPs were located on the same side as the AFMs. The terminal domains were located in the cytoplasmic region. For the Erp structure, DeepTMHMM does not predict transmembrane domains in the sequence and locates a signal peptide between residues M1 and P40.

For IDRs analysis, the results show high-order structures for RvPs with disordered terminal domains (Figure 1A,B). Despite the low-disorder probability values, the BDR analysis shows two regions with a higher probability of interacting with other structures. Both BDRs correspond to the cytosolic domain between residues M1-H15 and R72-R154. The highest ANCHOR2 values were mapped to the Rv1417 structure, with the residues K130-R154 being the most promiscuous region (Figure 1A). Analysis of Rv2617c showed only one BDR in the N-terminal domain (M1-K16) located in the cytosol region (Figure 1B).

A particular case in point is the Erp protein. Erp is associated with the bacterial cell wall; however, traces have been found in culture supernatants [23,24]. Furthermore, sequence analysis shows a transmembrane helical motif between residues G253 and V273 [26]. However, IDRs analysis shows a large part of the Erp structure with a high probability of being disordered with an average of 40% in transmembrane residues. Notably, the region between residues V50–V250 is recognized as a large BDR, suggesting a high interaction domain (Figure 1C).

Based on the above results, protein–membrane systems were constructed using AlphaFold models (AFMs) due to their well-defined membrane motifs: two alpha-helices in the structure of Rv1417 (W16-G36, T50-A71) and four in Rv2617c (P18-F43, A68-V88, P90-T110, and Y115-S140). Second modeling of the structure was performed to compare these models with those obtained with the I-TASSER algorithms. Figure 2A,C show the initial configuration of both proteins. As can be seen, the AFMs transmembrane helical structures are lost in the I-TASSER models (ITMs). The globular structures adopted in the ITMs are reflected in the initial radii of gyration of Rv1417 (1.22, 1.31, and 1.18 nm on the *x*, *y*, and *z* axis, respectively) and Rv2617c (1.10, 1.36, and 1.29 nm). Finally, the Erp protein was modeled as an unbound structure due to a large number of disordered residues. While AFMs consider it as an unfolded protein, ITMs constructed it as a compact protein.

### 3.2. Protein Structures at Different Medium Conditions

Initially, the three protein systems were prepared to evaluate the stability of the proteins (Rv1417 and Rv2617c) within their membranes, or under unrestricted conditions (Erp). For this purpose, 500 ns of MD simulations were performed using water molecules as solvent. Because the precise sites where protein–protein interactions occur have not been established, second MD simulations were performed to determine conformational stability under unrestricted membrane conditions. Considering that interactions could occur close to the membrane surface, a semi-polar environment was considered to mimic the interfacial region of the membrane [63,64]. For this purpose, explicit ethanol molecules (ϵ≈ 23.58 at 309.65K [37]) were used as solvent, and 200 ns of MD production was simulated.

#### 3.2.1. Rv1417 and Rv2617c Proteins

Figure 2B,D show the final configurations of both RvPs in membrane and unanchored conditions. As can be seen, although the AFMs structures are structurally well conserved, the transmembrane domain shows a small loss of helical structure. On the other hand, the unanchored structures present a drastic change in their conformations, losing secondary structures and compactness. Even, the α2 helix in the Rv1417 protein goes on to form a small anti-parallel β-sheet, which is a characteristic process of proteins to become disordered structures. In addition, several studies suggest that proteins capable of forming beta-sheets structures are also capable of self-assembly [65,66], which could explain the observation by Klepp et al. that both Rv1417 and Rv2617c could form homodimer complexes [24].

The predicted stability of the AFMs structures can be observed in the RMSD plots (Figure 3A). Both structures converge after 100 ns of the simulation, showing low values in the RMSD fluctuations (±0.05 for Rv1417 and ±0.06 nm for Rv2617c) compared to those obtained in the ITMs (±0.14 and ±0.11 nm, respectively) (Table 2). The interactions with membrane molecules allow them to retain their compactness with low fluctuation in their residues. Notably, the Rv1417 protein exhibits a well-defined external domain formed by both terminal domains (two anti-parallel beta-sheets and an alpha-helix). This domain is highly conserved across MD trajectories, which is reflected in the initial and final volume-area ratio (2.82 nm and 2.78 nm, respectively). In the case of the ITMs structures, a high loss of secondary structure changes the globular conformations to open structures. High fluctuations in the RMSD and RMSF plots, in addition to an increase in the radii of gyration, show a high instability, which is characteristic of disordered proteins. These results suggest that under semi-polar conditions (close to the membrane surface), RvPs can increase their promiscuous activity. This fact could explain why it is considered that protein–protein interactions could take place at the cell envelope [24].

Hydrogen bonding analyses show a similar number of intramolecular polar interactions in both AFMs (100/109) and ITMs structures (106/101) for Rv1417 and Rv2617c, respectively, at the beginning of the MD simulations. Despite the loss of compactness and secondary structure, the average hydrogen bond (HB) values of ITMs were comparable to those of the AFMs structures (Table 2). This fact can be explained by the decrease of polar sites due to ethanol molecules and the change in the electrostatic environment. However, it is interesting to note that under semi-polar conditions, the proteins change to unfolded forms while retaining their secondary structures. On the other hand, in AFMs structures, the interactions with the lipid membrane increase from initial values of 5 and 4 to mean values of 21 and 20 for Rv1417 and Rv2617c, respectively (Figure 3B). To identify the major residues involved in protein–membrane interactions, HB occupancies were calculated. Results show that in Rv1417 protein residues T50, R72, G89, and R108 were mainly involved in membrane interactions, whereas in Rv2617c, they were L11, R91, and R133.

By analyzing the MD trajectories, it could be observed that the proteins had some diffusion across the membrane (Figure 4A). Continuous movement is vital in physiological processes for the formation of protein–lipid and protein–protein complexes [67]. To evaluate the consistency of the protein–membrane systems and their ability to move across the lipid bilayer, density analyses of the components of the systems were performed, and the lateral diffusion of the proteins in the membranes was evaluated.

The density profiles show good membrane behavior throughout the MD simulations (Figure 4B). Taking the distance between the two peaks of the headgroups, which is directly related to the bilayer thickness, the membrane thicknesses were 4.2 and 4.1 nm for Rv1417 and Rv2617c systems, respectively. These results show a slightly higher thickness than those measured experimentally for pure membranes (3.0–4.0 nm) [68,69]. Protein density profiles (black line in Figure 4B) show the degree of embedding of the structures in the lipid bilayer. For Rv1417, the exposed region was 3.85 nm with a mean SASA value of 98.08 ± 2.13 nm${}^{2}$, whereas for Rv2617c, the values were 2.2 nm and 90.36 ± 2.14 nm${}^{2}$, respectively (Figure 4C,D).

Finally, the Gromacs tool *msd* was used to calculate the lateral diffusion of the proteins. The previous visual analysis of the protein trajectories showed the higher mobility of the Rv2617c protein across the lipid bilayer. However, mean square displacement (MSD) graph (Figure 4E) and diffusion coefficient values showed a higher lateral mobility of the Rv1417 protein (D = 7.86 × 10${}^{-9}$ cm${}^{2}$/s) with respect to Rv2617c (D = 5.62 × 10${}^{-9}$ cm${}^{2}$/s).

However, the displacement of membrane molecules was larger in Rv2617c than in Rv1417 (D = 5.20 × 10${}^{-7}$ and D = 4.86 × 10${}^{-7}$ cm${}^{2}$/s, respectively). These results and the fact that electrostatic interactions can affect diffusion in lipid membranes (both proteins showed similar HB interactions, Table 2) suggest a higher affinity of the Rv1417 protein to lipid bilayer molecules. Subsequent docking calculations were performed with this observation in mind.

#### 3.2.2. Polar Environment Affects the Compactness of Erp Structure

Using two unanchored isolated systems to characterize the structure of Erp under different polar conditions, simulation analyses showed remarkable results. As shown in Figure 5A,B, the initial Erp protein structures differ significantly in conformational shape. The AFMs structure shows an open unfolded shape (3.84 nm RG) with eight helical structures, four of them with more than 10 residues (R6-E35, G56-F69, V219-E228, and S232-Q251). In contrast, the ITMs structure has a globular shape (2.32 nm RG) with 15 small helical structures, the largest with only nine residues (P164-G172).

However, after MD simulations, AFMs adopts a closed conformation, while ITMs opens to an unstructured shape (2.04 ± 0.02 and 2.46 ± 0.07 nm of RG, respectively). The loss of helical structure is higher in AFMs (60.7%) compared to ITMs (32.9%), with the formation of a small anti-parallel beta-sheet observed between residues T210-T211 and G217-L218 in the latter structure. The same behavior was observed in the unanchored structures of Rv1417 and Rv2617c, showing a relevant effect of the medium on the interacting structures.

As mentioned above, disordered structures have a greater ability to interact with other structures, and this fact can be partially explained in Figure 5C. The vibrational motion of the residues is usually associated with the stability of the structures; however, it can also be associated with the ability to interact or bond with other structures. In this sense, the B-factor of Erp proteins was calculated, and the values were mapped onto the final MD structures, finding remarkable differences between both models. Taking the same scale of values, the AFMs shows only two residues with high vibrational values, P282 and V283 in contrast to the ITMs, which exhibit several high vibration regions in its structure.

### 3.3. Protein–Protein Complexes

To analyze protein–protein interactions, minimum energy structures were achieved through FEL analysis, and molecular docking calculations were performed to obtain the protein complexes. In total, 1000 protein complexes were analyzed for each heterodimer formed with the Erp protein, two for interactions with membrane-embedded RvPs, and two for unanchored systems in a semi-polar environment. All complexes were evaluated based on their binding energy obtained by a flexible refinement algorithm using the FireDock server. For complexes on a membrane, the 100 best structures were chosen to evaluate the interactions of residues in the contact zone. For the case of complexes in a semi-polar medium, the 20 best structures were analyzed.

#### 3.3.1. Membrane Systems

Figure 6 shows the steps used to obtain the main protein complexes involved in protein interactions. The protein dockings were performed taking into account the steric effect of the lipid bilayer. Thus, in several complexes, the Erp protein (used as a ligand) interacted only with membrane molecules (Figure 6A), and these complexes were discarded. In the case of the Rv1417–Erp complexes, there were 16 structures that made effective contact, 15 located in the cytoplasmic region and one in the extracellular region (Figure 6B). Residue contact analysis of the best solution for both regions is shown in Figure 6C. It is interesting to mention that the Erp protein in both complexes interacts with the C-terminal domain residues (Figure 6D), which apparently contradicts what was observed by Klepp et al. [24]. In their study, the authors mention that deletion of the C-terminal domain did not affect the association of Erp protein with RvPs, suggesting that this domain is not relevant in protein interactions. To address this question, an analysis of the contact frequencies of the 16 complexes was performed to determine the main residues involved in this coupling. The values were mapped as heat maps on the protein surfaces (Figure 6E,F).

Evaluation of the contacts at the protein interface shows a high interaction site in the N-terminal domain of the Rv1417 protein. This interaction hot spot is formed by residues M1-W8 and S79 (zoom in Figure 6E). Other important contact sites were located at residues A80, E97, D116, D117, R146-R148, and R154, which were all in the cytoplasmic region. As for the Erp protein, the heat map reaffirms the interaction of the C-terminal domain with the Rv1417 protein. The main hot spot was located at residue D221 and a minor contact site formed by residues A269–V273. However, several regions of interaction outside the C-terminal domain were also observed (Figure 6F, and Table 3).

Following the same methodology, the complexes obtained in the Rv2617c–Erp docking were analyzed. Despite having a smaller contact area with respect to the Rv1417 protein, Rv2617c interacted with the Erp protein in a greater number of complexes. Contact analysis shows 19 structures with protein–protein interactions, seven of them in the cytoplasmic region and 12 in the extracellular region of the membrane (Figure 7A). The analysis of the best solutions for both parts is shown in Figure 7B. The zoom of the cytoplasmic region shows a higher number of interacting residues in both proteins. Again, the C-terminal residues of the Erp protein were involved in the interactions, especially residues L242, P244, and S245 with several hydrophobic contacts with residues P5, T6, and T7 of Rv2617c. Contact frequency analysis shows three hot spots on the surface of Rv2617c: M1, S2, S140; T7, P9; and P140 in the cytoplasmic region. Importantly, although the extracellular region shows no hot spots, the entire surface shows residue interactions (Figure 7E).

The heat map of Erp interactions shows a higher affinity for the Rv2617c protein. Figure 7D shows several regions of residue contacts on the Erp surface. The major hot spots were located at residues T158, P159, and T180, and in the G188-G191 region. However, important contact regions were found to be H43-E44, G160, T192-P194, T213, and P244-S245. It is noteworthy that the heat map shows that interactions occur preferentially on one side of the Erp structure. Indeed, these interactions are present in both cytoplasmic and extracellular domains, which may increase the affinity between the two proteins compared to Erp–Rv1417 complexes (Table 3). These results may help explain recent experimental studies mentioning that Erp–Rv2617c complexes play a greater role in Mtb virulence and resistance to oxidative stress during host infections [23].

#### 3.3.2. Unanchored Complexes

Interactions between Erp protein and RvPs can take place at the cell envelope and under unanchored or free protein conditions [23,24]. In this work, we have been assumed to occur in regions close to the cell membrane Figure 8A. To understand the role of the electrostatic environment in Erp-RvPs interactions, the final structures obtained from semipolar MD simulations were docked. The protein complexes were analyzed under the same scheme used in the membrane analyses, and the main complexes for both Erp-RvPs systems are shown in Figure 8B,E.

The results show that all complexes exhibit augmented interaction at the contact interface due to the increased contact area. An example of this is shown in the zooms of the protein interfaces in both Erp–RvPs complexes, where large contact areas with more than 18 interacting residues are observed. It is interesting to note the fact that in the membrane complexes, all interactions found were of hydrophobic character, whereas in these complexes, electrostatic interactions appear (Table 3). The heat maps show a more uniform distribution of contacts on the protein surfaces. However, in the case of the Rv1417 protein, four well-defined interaction hot spots are observed, residue L63 and the areas between residues L68-R72, G81-L88, and Y119-R131 (Figure 8C). In the case of the Rv2617c protein, intermediate contact regions were detected between residues I3-T7, Y23-V32, T110-L119, and L134-A135 (Figure 8F). It is important to mention what was observed in the Erp proteins, since the heat maps do not show hot spots on their surface; i.e., the contacts are distributed in uniformly defined regions. Especially, interactions were observed in the repeat domains, specifically between residues L127 and T158. In addition, the C-terminal region also shows significant interacting residues, suggesting that this area could also form transient complexes with RvPs (Figure 8F).

Remarkable results were observed in the comparison of the binding energies (BEs) of the Erp–RvPs complexes. The Circos graphs show the binding energies of the first 1000 solutions obtained with the FireDock server (Figure 8G). The overall analysis shows a higher energetic affinity of the membrane complexes denoted by the blue–green hues of the graphs due to the higher amount of solutions with negative BEs. For the unanchored solutions, the predominant hue was red (positive BEs). Furthermore, Erp interactions with membrane molecules seem to stabilize the protein complexes and thus decrease the BEs. These results could indicate the preferred regions for the localization of protein interactions. Further analysis shows a higher energy affinity of the Erp protein for the Rv2617c protein under membrane-anchored conditions, as can be seen in the heat maps. However, under unanchored conditions, the same affinity was observed for both Erp–RvPs interactions due to the increased exposed area, as expected. The affinity of Erp for Rv2617c can also be observed in the contact histograms in the Circos plots, as the number of contacts was also higher for Erp–Rv2617c interactions in both membrane and unanchored conditions. These results seem to support the experimental observations regarding the higher relevance of Rv2617c in Erp [23] functions.

Finally, a comparison of the binding energy of the best complexes shows a higher energetic affinity of the unanchored structures, mainly in the Erp–Rv1417 complexes, which were 27.8% more energetically favorable as opposed to 11.9% for the Erp–Rv2617c complexes (Figure 8H). Even several Erp–Rv2617c membrane complexes were energetically more favorable. These results suggest that despite the lower probability of interactions taking place at the lipid bilayer surface, the heterodimers formed will have higher coupling energy.

#### 3.3.3. The Erp Protein Is the Main Guideline for Protein Interactions

To clarify the nature of protein interactions, an ESP analysis was carried out to correlate the energy affinity with the electrostatic properties of the proteins. For this purpose, all structures were subjected to hydrogen optimization to avoid overlaps and to improve the hydrogen-bonding networks in the protein structures [62,70]. Two ESP calculations were generated for each complex, the first taking each protein separately (unbound complex) and the second taking the Erp protein as ligand and attached in RvPs structures (bound complex). The charges used for Erp were those obtained in the Gromacs topologies. For membrane complexes, both the cytoplasmic and extracellular regions were analyzed by taking a polar electrostatic environment with a dielectric constant of 79.0. For unanchored complexes, the same dielectric constant of the MD calculations was used (23.58).

The results show remarkable differences in protein interactions due to the electrostatic effect in both environments (Figure 9). The most relevant electrostatic feature is observed on the Erp surface. In both environments, Erp presents larger negative regions; however, in a semi-polar medium, this electrostatic character is increased. This fact is reflected in its interactions with the RvPs, which are markedly different. Figure 9A,B show the interactions and contact surfaces for the solutions analyzed in the membrane systems. As can be seen, protein–protein interactions have a local impact on the ESP surfaces in both structures. In fact, only residues with effective interactions change their electrostatic character. It is noteworthy that the protein interactions in the polar medium had a hydrophobic character denoted by the white hues on the non-interacting structures (Figure 9A,B). In contrast, when the structures form the heterodimers, the contact interfaces change to a hydrophilic character denoted by the red and hues.

On the other hand, in a semi-polar medium, drastic changes in the ESP surfaces were observed. As can be seen in Figure 9C,D, the increase in the electrophilic character of both structures has a significant impact on the ESP of the interfaces. In the case of the Erp surfaces, the negatively charged regions were so intense that the interaction with the RvPs did not significantly modify their ESP surface. Nevertheless, when looking at the Rv protein surfaces, they drastically change their hydrophilic character from a positive to a negative surface, indicating the electrostatic nature of protein interactions in a semi-polar environment. These results show an increased influence of the Erp protein on the interactions with RvPs and suggest that the formation of heterodimers is highly dependent on this protein and its ability to interact with RvPs structures.

## 4. Conclusions

The search for new therapeutic targets to counteract the propagation of the bacterium *Mycobacterium tuberculosis*, which causes tuberculosis disease, has been increasing. Although the virulence factor Erp protein has been demonstrated in several experimental studies, its function and molecular interactions with other proteins have been poorly addressed. In this work, using MD simulations, docking calculations, and various in silico techniques, we have explored the molecular interactions of the Erp protein with two putative proteins identified by Klepp et al., Rv1417 and Rv2617c. To understand the importance of the medium in Erp–RvPs interactions, two dielectric conditions, a polar solvent with water molecules and a semi-polar solvent with ethanol molecules, were studied.

Our results show several highlights in the molecular characterization of Erp–RvPs interactions. The structural characterization of the proteins showed that both RvPs possess domains characteristic of transmembrane proteins. The low probability of having intrinsically disordered regions corroborated this feature. Contrary to previous studies, the N- and C-terminal regions were found to be located in the cytoplasmic area of the cell. These results coincide with the structures obtained by the AlphaFold methodology. MD simulations showed that under polar conditions, the proteins adopt a closed globular conformation, whereas under semi-polar conditions, open structures were observed. These open forms allowed the proteins to increase their interaction capacity by increasing their contact area. In addition, protein promiscuity could be favored by the high vibrational motion of the residues in the unfolding process. Mean square displacement analysis showed a higher affinity of the Rv1417 protein with membrane molecules. This affinity, whose effect is a smaller translation of the protein, could indicate that in the formation of the RvPs heterodimers, this protein would act as a receptor of the complexes.

Important results were obtained when performing the molecular dockings between the RvPs and the Erp protein. When the binding energy of the docking calculations was analyzed, we could observe that the energetic affinity was higher in membrane-bound systems than in unanchored ones. Furthermore, about 90% of the solutions obtained in unanchored systems had positive interaction energies. It is noteworthy that the contact analyses showed that all residue contacts in the membrane systems were hydrophobic in nature. Moreover, we observed that the C-terminal residues of the Erp protein were involved in most of the complexes analyzed, forming hot spots in the contact frequency analysis. This latter result suggests the possibility of the formation of transient complexes involving the C-terminal domain. Interactions were also found in the Erp repeat domains, which could confirm that these regions are involved in interactions with RvPs. An important result was to corroborate the higher energetic affinity between the Erp and Rv2617c protein under membrane conditions, which is in agreement with experimental studies.

Finally, ESP analysis showed the importance of the dielectric conditions of the medium in protein interactions and confirmed the directionality of the Erp protein in complex formation. In the case of polar conditions, the protein interactions had a hydrophobic character denoted by the white hues in both structures. However, under semi-polar conditions, the electrostatic nature of proteins changes dramatically, causing the interactions to be hydrophilic. The latter is of great relevance, since it has been observed that the electrostatic environment has a low dielectric constant in the areas close to the cell membrane.

Several limitations are observed in our study. However, it opens different perspectives regarding the analysis of one of the main virulence factors of MTB, such as the Erp protein, from a molecular point of view. It is suggested for future work the analysis of the free binding energies of protein complexes as well as the study of drugs that can inhibit these interactions. Overall, this work aims to provide novel insights into the study and elucidation of the molecular interactions between Erp and Rv proteins.

## Figures and Tables

**Figure 1 polymers-14-02577-f001:**
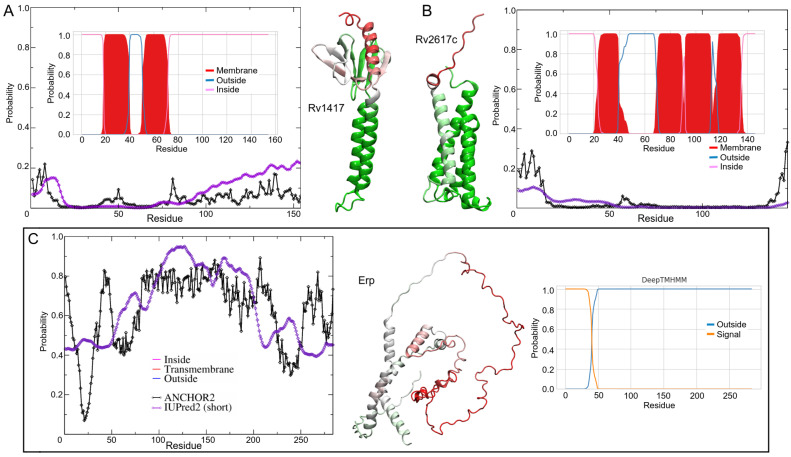
Prediction of transmembrane helices and disordered protein regions of the Mtb proteins studied. The inner plots show the probability of residue forming helical structures. The probability was calculated based on a deep learning protein language model-based algorithm. The filled curves (red color) are the helical residues, while blue and magenta lines mean the inside or outside membrane regions, respectively. The disorder regions (outer plots) were analyzed by IUPred2 (purple line) and ANCHOR2 (black line) algorithms. For a better representation, the disorder probability values were mapped on the 3D structures. Red, white, and green colors represent high, mean, and low probabilities of disordered conformation, respectively. (**A**) Rv1417, (**B**) Rv2617c, and (**C**) Erp.

**Figure 2 polymers-14-02577-f002:**
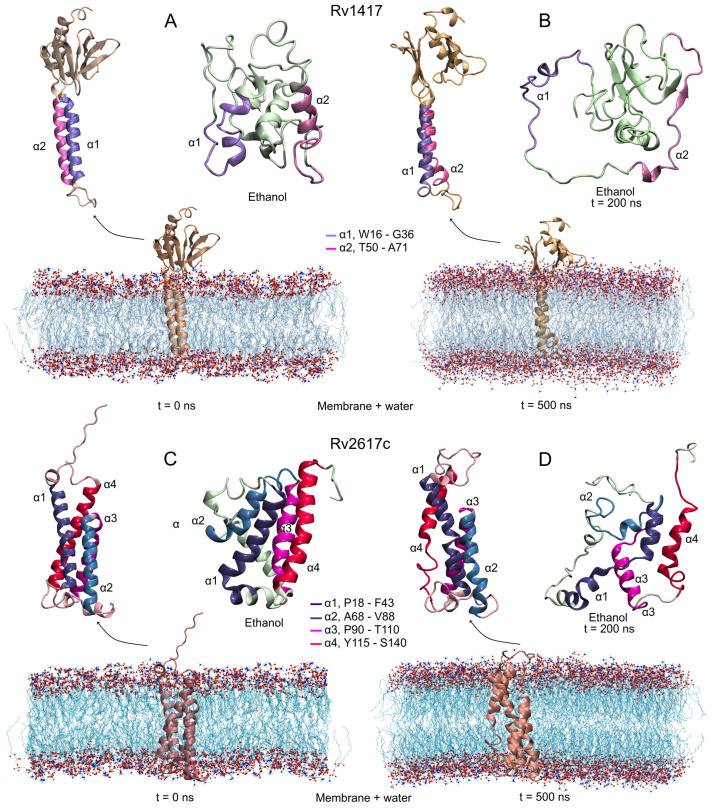
Initial and final structures obtained in the MD simulations. Protein systems were simulated under two different conditions: protein–membrane in water solvent and in semi-polar (ethanol as solvent) environments. (**A**,**B**) Rv1417, and (**C**,**D**) Rv2617c. Transmembrane alpha-helices are highlighted in all structures. MD simulations were carried out at 309.15 K and 1 bar conditions.

**Figure 3 polymers-14-02577-f003:**
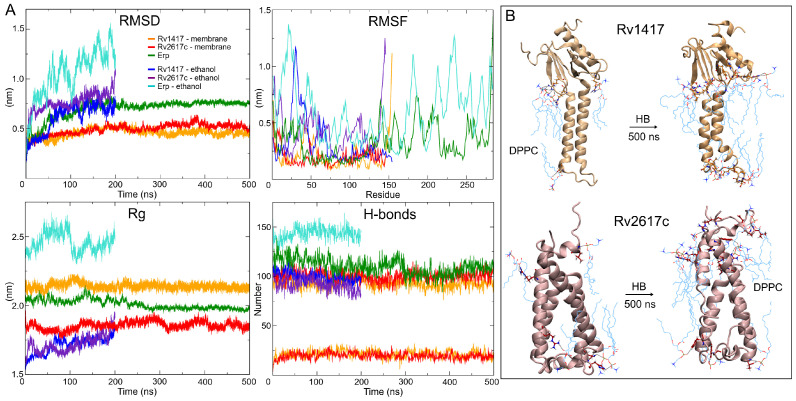
MD results. (**A**) Stability descriptors of the protein systems. (**B**) Intermolecular hydrogen bonds found at the initial and final frames of MD trajectories. The DPPC membrane molecules are depicted by cyan lines and protein residues by CPK representation.

**Figure 4 polymers-14-02577-f004:**
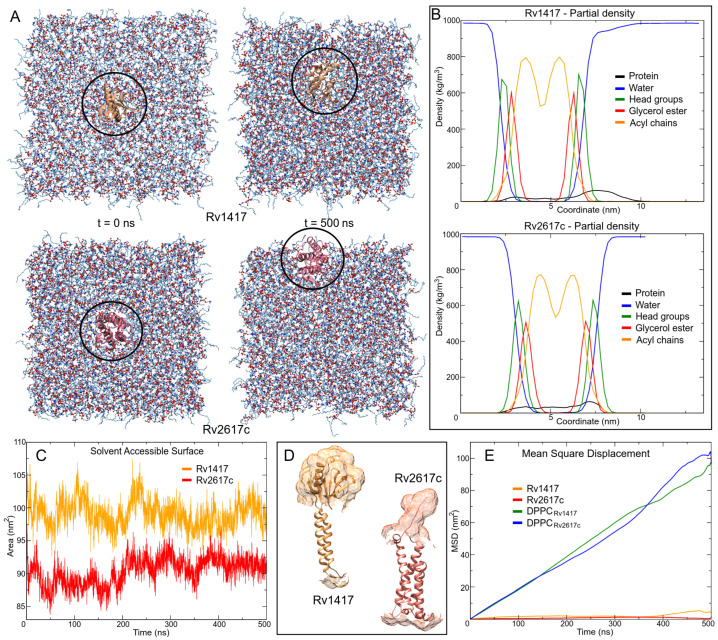
Protein–membrane systems. (**A**) Movement of proteins through the lipid bilayer. (**B**) Density profiles of system components along the *z*-axis. (**C**) Solvent accessible surface area (SASA) calculation. (**D**) Connolly surface of the outer protein regions. (**E**) Mean square displacement (MSD) calculations were used to measure the lateral diffusion of protein and lipid head groups. MSD values of lipids are based on the diffusion of DPPC headgroup P8 atoms.

**Figure 5 polymers-14-02577-f005:**
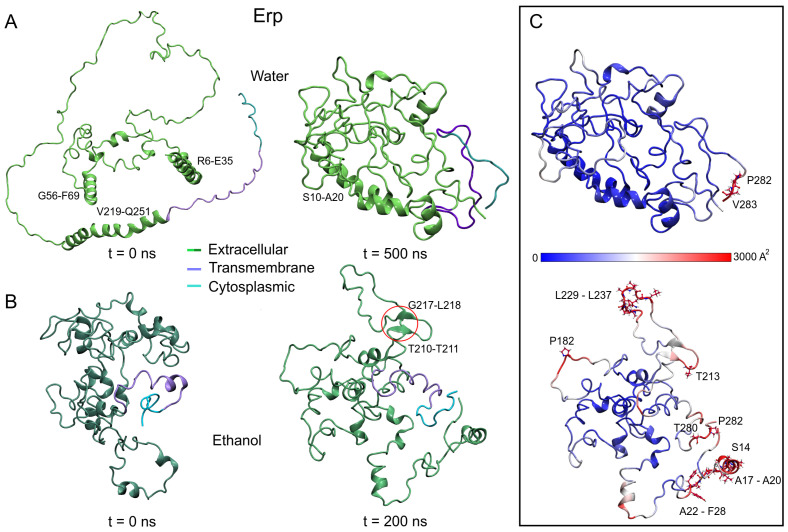
Erp structures at the initial and final of the MD trajectories. (**A**) Model retrieved of AlphaFold server. (**B**) Model built using the I-TASSER server. (**C**) B-factor values mapped on final MD structures. The high vibrational residues were depicted as a ball and stick and were labeled black. Since the Erp protein has intrinsically disordered domains, the size of the simulation boxes was revised to avoid interactions of the protein with its periodic images. Main domains are highlighted.

**Figure 6 polymers-14-02577-f006:**
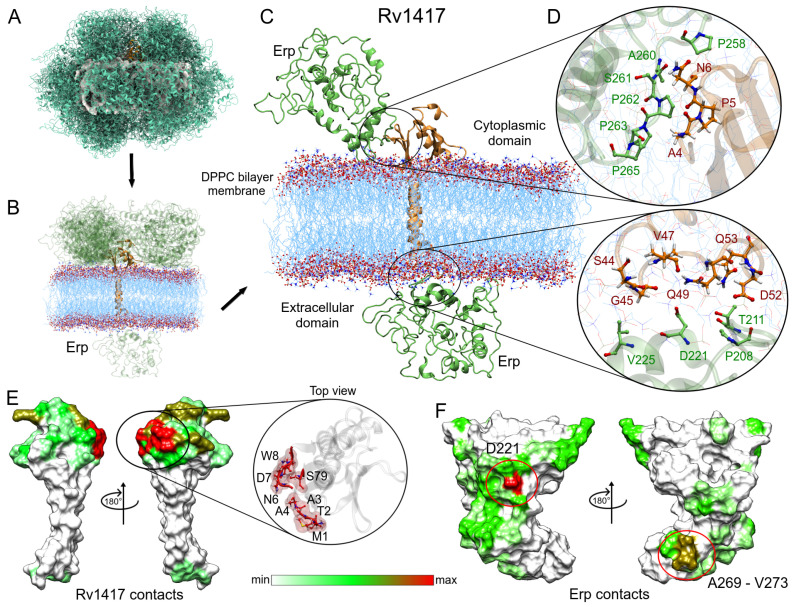
Analysis of the Erp–Rv1417 interactions. (**A**) Overall representation of the 100 energetically most favorable molecular complexes obtained in FireDock calculation. (**B**) Best complexes with effective protein contact. The up-side corresponds to the cytoplasmic environment, while the down-side to the extracellular domain. (**C**) Contact residues on the protein interface. Erp and Rv1417 residues are depicted in green and red–orange colors, respectively. (**D**) Zooms of the interface interactions. (**E**,**F**) Heat map of the main contacts between the Rv1417 protein and Erp protein. Red, green and white colors indicate a high or a medium number of residue contacts, or no contacts at all, respectively.

**Figure 7 polymers-14-02577-f007:**
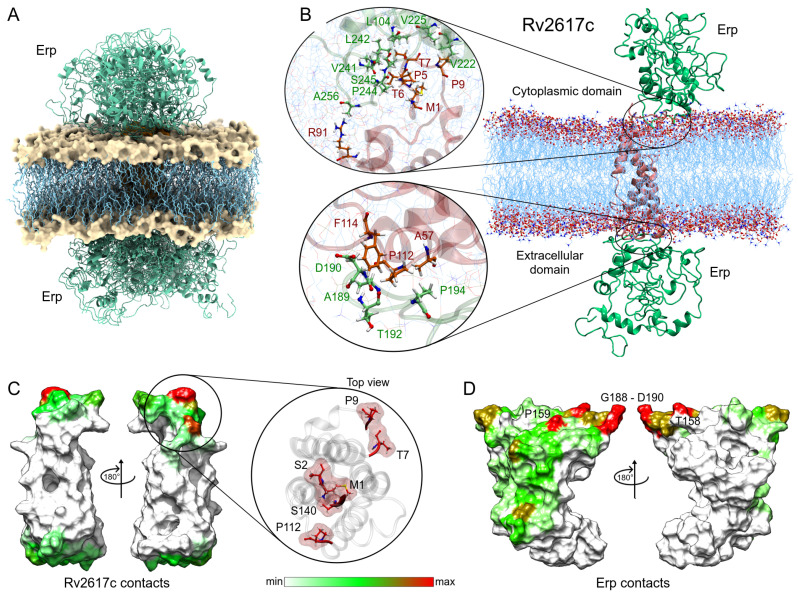
Analysis of the Erp–Rv2617c interactions. (**A**) Overall representation of the 19 energetically most favorable molecular complexes obtained in FireDock calculation with effective protein contact. (**B**) Contact analysis of the two energy best solutions on both sides of the lipid bilayer. Erp and Rv2617c residues are depicted in green and red–orange colors, respectively. (**C**,**D**) Heat map of the main contacts between the Rv2617c protein and Erp protein. The color scheme is the same as that in Figure 6E,F.

**Figure 8 polymers-14-02577-f008:**
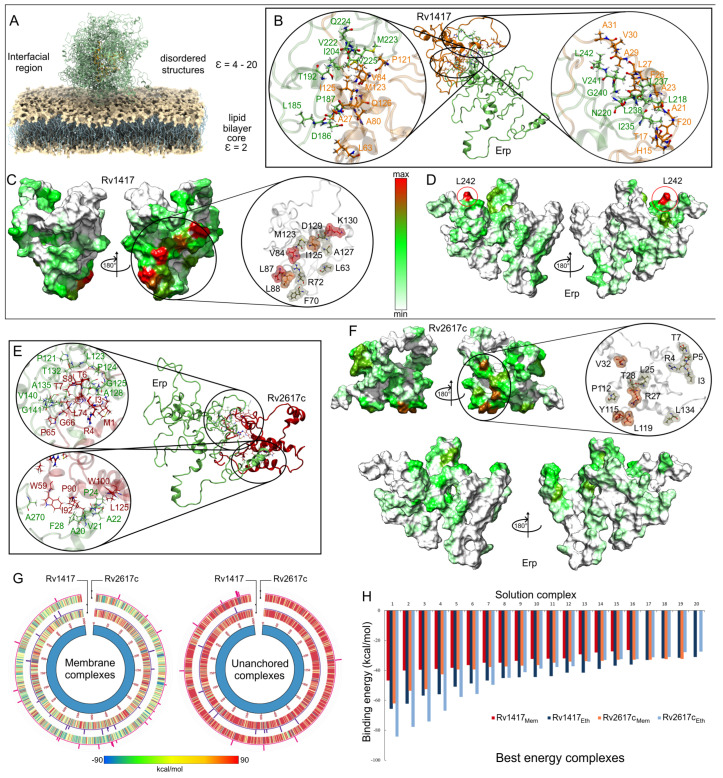
Contact analysis of the unanchored heterodimers under semi-polar conditions. (**A**) Representation of the location of the Erp–RvPs complexes. The dielectric constant value on the environment on the membrane surface was considered close to 20, which is an approximate value to that of the ethanol at 309.65 K (23.58). (**B**,**E**) Interactions on the protein interface for the best energy complexes in both Erp–RvPs couplings. The same color scheme was used to depict the protein residues of the previous figures. (**C**,**D**,**F**) Heat maps of the contact frequencies mapped on the final structures obtained in the MD semi-polar simulations. (**G**) Energy comparison of the 1000 best-ranked Erp–RvPs complexes under both the membrane and semi-polar conditions. Heat maps correspond to the binding energies by complex, while histograms represent the final solutions used in contact analyses, and their bars height is the number of contacts found in a maximum of 30 contacts. (**H**) Binding energies of the 20 solutions used in the contact analyses.

**Figure 9 polymers-14-02577-f009:**
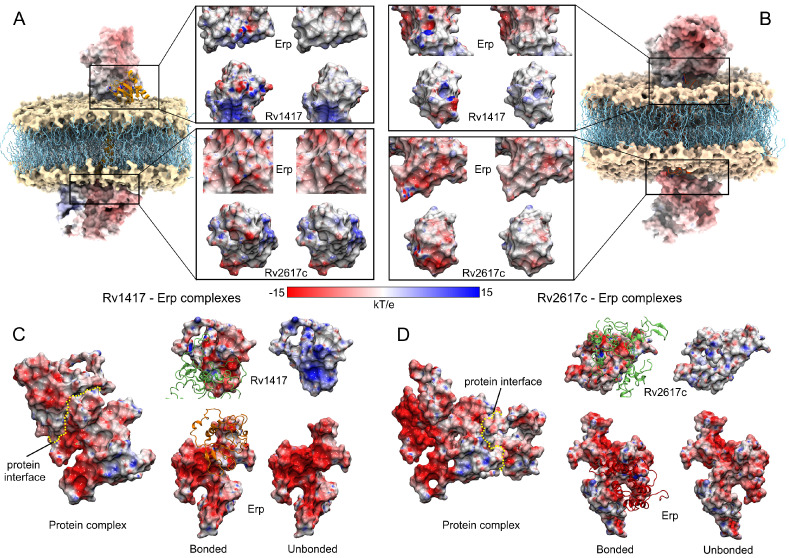
Electrostatic analysis of the best protein complexes obtained in the docking calculations. (**A**,**C**) Rv1417–Erp complexes. (**B**,**D**) Rv2617c–Erp complexes. For membrane systems, two solutions were analyzed in the cytoplasmic and extracellular regions. The insets in A and B show the contact surfaces between the Erp–RvPs structures. Within these insets, figures on the left represent the protein surfaces when they are attached, while figures on the right are the non-interacting structures. For unanchored complexes (**C**,**D**), the yellow dotted line represents the interface between the proteins. Contact surfaces are shown in the front view. For both membrane and unanchored systems, the same color scale was used. Blue, red, and white colors represent positively charged, negatively charged, and neutral surfaces, respectively.

**Table 1 polymers-14-02577-t001:** MD simulation conditions used in this work.

Parameter	Membrane Systems	Semi-Polar Systems
NVT equilibrium		
Constraints	all bonds	h-bonds
Cutoff ${}^{a}$	1.2 nm	1.1 nm
T${}_{coupl}$	V-rescale	V-rescale
Temperature	323.15 K	309.65 K
$\tau_{T}$	0.1 ps	0.5 ps
NPT equilibrium		
Constraints	all bonds	h-bonds
Cutoff ${}^{a}$	1.2 nm	1.1 nm
T${}_{coupl}$	Nose–Hoover	V-rescale
Temperature	323.15 K	309.65 K
$\tau_{T}$	0.1 ps	0.1 ps
P${}_{coupl}$	Parrinello–Rahman	Parrinello–Rahman
P${}_{coupl}$ type	semi-isotropic	isotropic
Pressure	1.0 bar	1.0 bar
$\tau_{P}$	5.0 ps	2.0 ps
Production phase		
Constraints	h-bonds	h-bonds
Cutoff ${}^{a}$	1.2 nm	1.2 nm
T${}_{coupl}$	Nose-Hoover	Nose-Hoover
Temperature	309.65 K	309.65 K
$\tau_{T}$	0.5 ps	0.1 ps
P${}_{coupl}$	Parrinello–Rahman	Parrinello–Rahman
P${}_{coupl}$ type	semi-isotropic	isotropic
Pressure	1.0 bar	1.0 bar
$\tau_{P}$	2.0 ps	1.0 ps
Time trajectory	500 ns	200 ns

^*a*^ Same cutoff value for electrostatic, van der Waals and rlist was used.

**Table 2 polymers-14-02577-t002:** Stability descriptors of the *Mtb* proteins.

System	RMSD ${}^{a}$	RMSF ${}^{a}$	RG ${}^{a}$				H-Bonds	H-Bonds
			Total	*x* Axis	*y* Axis	*z* Axis		+ DPPC
Membrane and water—500 ns								
Rv1417	0.46 ± 0.05	0.22 ± 0.14	2.14 ± 0.02	2.03 ± 0.03	2.01 ± 0.03	1.00 ± 0.05	93 ± 5	21 ± 4
Rv2617c	0.50 ± 0.06	0.20 ± 0.07	1.85 ± 0.03	1.73 ± 0.04	1.71 ± 0.03	0.98 ± 0.04	100 ± 5	20 ± 3
Erp	0.65 ± 0.12	0.33 ± 0.17	2.04 ± 0.02	1.72 ± 0.12	1.55 ± 0.15	1.70 ± 0.14	114 ± 8	-
Ethanol—200 ns								
Rv1417	0.63 ± 0.14	0.32 ± 0.21	1.71 ± 0.06	1.41 ± 0.09	1.43 ± 0.04	1.36 ± 0.05	100 ± 5	-
Rv2617c	0.77 ± 0.11	0.44 ± 0.19	1.72 ± 0.06	1.24 ± 0.06	1.51 ± 0.06	1.44 ± 0.07	92 ± 7	-
Erp	1.05 ± 0.25	0.60 ± 0.26	2.46 ± 0.07	2.04 ± 0.09	1.90 ± 0.06	2.07 ± 0.10	143 ± 7	-

^*a*^ values are given in nanometers.

**Table 3 polymers-14-02577-t003:** Main contact residues in heterodimer complexes.

	**Rv1417**	**Erp**	**Rv2617c**	**Erp**
**Membrane systems**	M1, T2, A3, A4, N6, D7, W8, S79, A80, E97, D116, D117, R146, Y147, R148, R154	D221, A269, A270, A271, A272 V273, P274, P275	M1, S2, P5, T6, T7, P9, Q50, N53, M54, A57, D62, T67, A68, G111, P112, F114, S140, G141, R145, P146	H43, E44, T158, P159, G160, T180, G188, A189, D190, G191, T192, Y193, P194, T213, P244, S245
	**Rv1417**	**Erp**	**Rv2617c**	**Erp**
**Unanchored systems**	L27, **A29**, **H32**, A34, **G35**, G36, L37, L38, **K40**, **A55**, M56, **L59**, L63, **A66**, L68, F70, R72, **R76**, **G81**, **S83**, V84, **N86**, **L87**, L88, **D90**, **F103**, **Y119**, P121, M123, I125, Q126, A127, V128, D129, **K130**, D131, **R146**	**T11**, **T34**, **E44**, **T80**, **A85**, **P107**, **A145**, T147, **A150**, L151, P154, T158, **D186**, **Y193**, P194, I195, L196, P208, T211, **T213**, **G216**, **N227**, **V241**, L242, M243, P244, **M247**, **Q251**, **P258**, **A260**, P265	I3, **R4**, P5, **T6**, T7, Q14, **Y20**, **Y23**, V24, L25, R27, T28, F30, V32, M72, L74, **A77**, **T110**, P112, G113, F114, Y115, D116, **A118**, L119, L124, **L134**, A135, **H139**	A20, V21, **S23**, P24, C25, F28, **L58**, **F69**, L127, **A128**, A135, P138, V140, G141, **D186**, G202, S206, **S212**, **N227**, M243, **Q248**, **Q251**, **N252**, A255, **A256**, P262, A269, P274

Bold letters indicate electrostatic interactions.

## Data Availability

Not applicable.

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
