# Peer review of "Novel In Silico Insights into Rv1417 and Rv2617c as Potential Protein Targets: The Importance of the Medium on the Structural Interactions with Exported Repetitive Protein (Erp) of Mycobacterium tuberculosis"

_polymers, 2022, doi:10.3390/polym14132577_

Round 1

Reviewer 1 Report

This manuscript could be accepted for publication in Polymers. The topic is important for medical- and bio-chemistry. Nowadays, tuberculosis is the second leading cause of death from a monopathogenic transmitted disease, only ahead of COVID-19. The role of exported repetitive protein (Erp) in the virulence of Mycobacterium tuberculosis has been extensively demonstrated. In vitro and in vivo assays have identified that Erp interacts with Rv1417 and Rv2617c proteins forming putative transient molecular complexes prior to localization to the cell envelope. Although new insights into the interactions and functions of Erp have emerged over the years, knowledge about its structure and protein-protein interactions at the atomistic level has not been sufficiently explored. In this work, authors have combined several in silico methodologies to gain new insights into the structural relationship between these proteins. Two system conditions were evaluated by MD simulations: Rv1417 and Rv2617c embedded in a lipid membrane and another with a semi-polar solvent to mimic the electrostatic conditions on the membrane surface. The Erp protein was simulated as an unanchored structure. Stabilized structures were docked and complexes were evaluated to recognize the main residues involved in protein-protein interactions. Results show the influence of the medium on the structural conformation of proteins. Globular conformations were favored under high polarity conditions and showed a higher energetic affinity in complex formation. While disordered conformations were favored under semi-polar conditions and an increase in the number of contacts between residues was observed. In addition, the electrostatic potential analysis showed remarkable changes in protein interactions due to the polarity of the medium, demonstrating the relevance of Erp protein in heterodimer formation. On the other hand, contact analysis showed that several C-terminal residues of Erp were involved in the protein interactions, which seems to contradict experimental observations, however, these complexes could be transient forms. The findings presented in this work are intended to open new perspectives in the studies of Erp protein molecular interactions and to improve the knowledge about its function and role in the virulence of Mycobacterium tuberculosis. The introduction provide sufficient background. The research methodology is adequate (may be, it would be a good idea to cite some additional papers to show that other researchers also utilized similar approaches for MD simulations [Symmetry 2021 13 (7), 1119; ACS omega 2021 6 (27), 17267-17275] and electrostatic surface potential calculations [CrystEngComm 2019 21 (4), 616-628]; Chemistry-A European Journal 2019 25 (36), 8590-8598; Chemistry–An Asian Journal 2019 14 (21), 3915-3920). The results are clearly presented. The conclusions supported by the data. The manuscript good illustrated and interesting to read. The overall style is very fine.

Author Response

The review of the manuscript by the referees is appreciated. The references suggested were reviewed, and although they touch on molecular simulation, we believe they are not in the direction of what we intend to show in this manuscript.

Reviewer 2 Report

The authors present a computational study about the interaction among ERP protein and Rv1417 and RV2617c proteins from M. tuberculosis. In general, the work will be of interest to those working with in silico tools but specially to those involved in antibacterial drug design. There are some points to be considered.

1.- The quality of some figures needs to be improved because insets are difficult to read, for example, figures 1, 3, and 4.

2.- Along the manuscript all the data are described in detail but, at the end, the most important point about them is not stated clearly.

3.- In the same context, a deeper discussion of the results would be helpful.

4.- Why was not performed the same simulation time in the both conditions assessed 500 ns vs 200 ns?

Author Response

The review of the manuscript by the referees is appreciated. Next, we respond to the four observations made:

Point #1: The quality of some figures needs to be improved because insets are difficult to read, for example, figures 1, 3, and 4.

Response 1. We thank the reviewer for his comments. The figures to which the reviewer refers have been improved, and the font size has been increased for easy reading. All recommendations were addressed and corrected.

Point #2: Along with the manuscript, all the data are described in detail but, at the end, the most important point about them is not stated clearly.

Response 2. In response to the reviewer's recommendations, we have added some lines that we consider relevant to clarify the conclusions of this work (lines 474-479, 483-489, 491-493, and 501-508). In addition, we have divided each of the ideas that we wanted to address in this section into paragraphs.

Point #3: In the same context, a deeper discussion of the results would be helpful.

Response 3. We appreciate the reviewer's suggestion, again. Although there are still issues to be addressed, we believe that these would be better understood if alternative simulations are considered, which we are already preparing for a future study. On the other hand, we also think that delving deeper into the discussions could make it difficult to read and understand this work. 

Point #4: Why was not performed the same simulation time in both conditions assessed 500 ns vs 200 ns?

Response 4. This question is an excellent observation from the reviewer. The reason is simple but implies a future line of research in our group. Under low dielectric constant conditions, the proteins we worked on began to unfold, making the simulation cells unsuitable for simulations. That is, we started to have problems with the periodicity of the systems, and the readings we had (at simulation times greater than 200 ns) were erroneous. However, we plan to address this issue in future studies.
